# Harnessing Spectrum Video for Subject-Level Few-Shot and Cross-Montage EEG Generalization

**Wei Wang** [1 2] **Fang He** [1 3] **Yifan Li** [1 3] **Wanying Qu** [3] **Yawei Li** [4 5] **Quanying Liu** [2 6] **Yanwei Fu** [1 3 7]

## Abstract

Existing EEG models are limited by electrode heterogeneity and rigid "channel-first" architectures that treat sensors as independent features. We propose Brain Signal Rendering (BSR), which reinterprets EEG as a physical projection of neural activity and transforms raw signals into structured spatiotemporal tensors (termed Spectrum Videos), enabling the transfer of rich priors from video foundation models. By utilizing VideoMAE for self-supervised pre-training, BSR learns robust, layout-agnostic spatiotemporal representations that preserve neural topology. We further employ subject-level few-shot learning and introduce cross-montage fine-tuning to rigorously evaluate generalization across subjects and electrode configurations. Experiments show that VideoMAE model integrated with the BSR framework significantly outperforms state-of-the-art spectrum based methods, providing a scalable and data-efficient foundation for generalizable EEG modeling. Our code is available at https://github.com/yanweifu-sii/BSR-VideoMAE.

## 1. Introduction

Electroencephalography (EEG) provides one of the most accessible and information-rich windows into human brain activity, enabling sustained progress in seizure detection (Shoeb & Guttag, 2010; Chen et al., 2026; Tegon et al., 2025), motor imagery decoding (Ma et al., 2022), emotion recognition (Duan et al., 2013; Zheng & Lu, 2015; Zhang et al., 2025), EEG-to-fMRI synthesis (Li et al., 2024b; Qu et al., 2026), visual reconstruction (Li et al., 2024a; Gao

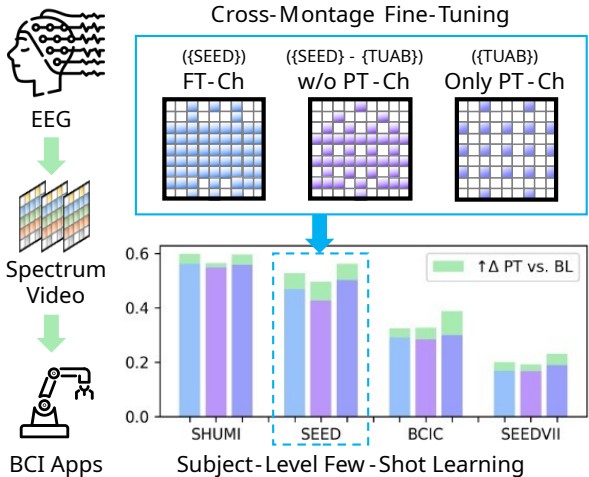

*Figure 1.* Overview of the proposed framework for robust EEG generalization. Transforming EEG into spectrum video enables unified representation across heterogeneous datasets. We evaluate cross-montage fine-tuning by measuring pre-training gains (PT vs. BL) on the full channel set (FT-Ch), seen channels (Only PT-Ch), and entirely novel channels (w/o PT-Ch). Additionally, subject-level few-shot learning benchmarks rapid adaptation by treating each subject as a distinct task.

et al., 2025), and effective connectivity estimation (Kiebel et al., 2008; Yang et al., 2023b). Despite decades of methodological advances, the scalability and generalization of EEG learning systems remain fundamentally limited by three challenges.

First, *severe heterogeneity in electrode layouts across datasets and acquisition protocols* introduces pervasive channel mismatch. EEG recordings differ widely in electrode count (e.g., 16–256 channels), spatial placement, referencing schemes, and missing or noisy sensors. As a result, the same channel index may correspond to different anatomical locations, or be entirely absent across datasets, breaking the implicit assumption of channel-wise correspondence that underlies most existing models and severely degrading cross-dataset generalization.

Second, the lack of a *unified EEG representation* that abstracts away sensor-specific configurations remains unresolved. An ideal representation should be invariant to elec-

[1]Shanghai Innovation Institute [2]Southern University of Science and Technology [3]Fudan University [4]ETH Zürich [5]Nanyang Technological University [6]Shenzhen Loop Area Institute [7]Zhejiang Normal University. Correspondence to: Yanwei Fu <yanweifu@fudan.edu.cn>, Quanying Liu <liuqy@sustech.edu.cn>.

*Proceedings of the 43rd International Conference on Machine Learning*, Seoul, South Korea. PMLR 306, 2026. Copyright 2026 by the author(s).

trode layouts, robust to missing channels, and transferable across subjects and tasks, while preserving fine-grained spatiotemporal neural dynamics. Such a representation is crucial for rapid adaptation in *low-resource and subject-specific* settings, yet remains largely unexplored.

Third, *the prevailing architectural bias of channel-first EEG models* fundamentally misrepresents the nature of EEG signals. Recent deep learning advances, from task-specific architectures (Jing et al., 2023) to large-scale EEG foundation models (Yang et al., 2023a; Jiang et al., 2024; Wang et al., 2024; 2025; Zhou et al., 2026), have significantly improved representation learning. However, these models largely adopt rigid, channel-indexed formulations that treat electrodes as independent and semantically aligned features. To address this, Ouahidi et al. (2025) proposes an innovative approach by embedding electrodes directly from spatial coordinates. While this work highlights the critical need for flexible sensor representation, a more comprehensive validation paradigm is still required to systematically assess channel generalization across the field. In reality, EEG channels are samples from **a spatially structured and continuous sensor array**, whose geometry varies across devices and acquisition settings. This inductive bias mismatch limits few-shot adaptation, amplifies sensitivity to channel heterogeneity, and renders existing evaluation protocols poorly aligned with real-world EEG deployment.

This paper tackles these three challenges by introducing a new perspective: treating EEG as a *physical projection* rather than a flat feature vector. We model EEG as the outcome of a *physical measurement process*, where electrodes form a two-dimensional sensor array that captures latent neural activity evolving in 3D space over time. Under this view, electrode heterogeneity corresponds to changes in projection geometry—analogous to different camera viewpoints of the same scene—rather than feature mismatch. Consequently, EEG representation learning is reframed as recovering shared spatiotemporal neural dynamics, rather than fitting task-specific embeddings tied to fixed channel layouts.

The core of our approach is rendering EEG signals as a *Spectrum Video* as in Fig. 1. Particularly, motivated by the physical projection perspective, we propose *Brain Signal Rendering (BSR)*, which transforms EEG spectra into a geometry-aware spatiotemporal representation. BSR spatializes electrodes according to their physical coordinates and encodes spectral responses as a dynamic image sequence, explicitly preserving neural topology (Figure 2). This unified spatiotemporal representation captures both spectral content and electrode geometry, enabling the transfer of learned spatiotemporal priors from pre-trained video foundation models like VideoMAE (Tong et al., 2022). Crucially, VideoMAE's self-supervised pre-training paradigm is partic-

ularly well-suited for learning robust representations amidst the significant variability across EEG datasets, while its spatiotemporal inductive biases naturally align with neural dynamics.

We further introduce a novel evaluation framework that goes *beyond fixed montages*. Typically, existing EEG foundation model evaluations rely on full-dataset fine-tuning with fixed electrode layouts, offering little insight into robustness under subject shift or montage variability. To address this, we propose two realistic and challenging settings: (i) *subject-level few-shot learning*, which adapts the cross-subject transfer learning protocol (Li et al., 2026) to treat each subject as a separate task requiring rapid adaptation from limited data—to our knowledge the first application of this paradigm for probing EEG foundation model generalization; and (ii) *cross-montage fine-tuning*, introduced here for the first time, which assesses transfer across datasets with different electrode configurations.

Formally, we propose a three-stage framework for EEG representation learning. First, *Brain Signal Rendering (BSR)* converts raw EEG into a *Spectrum Video* by extracting time–frequency spectra and spatializing them according to electrode coordinates, preserving both spectral and spatial neural dynamics. Second, the rendered videos are processed by a video encoder (VideoMAE) that undergoes self-supervised pre-training on spectrum videos; this process allows the model to bridge the domain gap between natural videos and neural signals by capturing the underlying spatiotemporal regularities of EEG. Finally, the encoder is fine-tuned under realistic transfer settings, including *subject-level few-shot learning* and *cross-montage fine-tuning*, enabling robust adaptation across subjects, tasks, and electrode configurations. Critically, subject-level few-shot learning reflects real-world deployment by requiring adaptation from only a few sessions per subject, handling variability in hardware, protocols, and individual physiology. Additionally, *cross-montage fine-tuning* introduces a more rigorous evaluation paradigm for channel generalization by assessing performance on both seen and previously unseen electrode subsets. Extensive experiments demonstrate that our framework consistently improves generalization and achieves substantial performance gains, providing a scalable, interpretable, and data-efficient foundation for EEG modeling.

**Contributions.** (1) We introduce **Brain Signal Rendering**, which transforms raw EEG into geometry-aware spatiotemporal videos compatible with VideoMAE. By leveraging a **self-supervised pre-training** paradigm across heterogeneous datasets, the video encoder learns robust, layout-agnostic spatial-temporal representations. (2) We propose **cross-montage fine-tuning**, a rigorous paradigm that systematically evaluates channel generalization by benchmark-

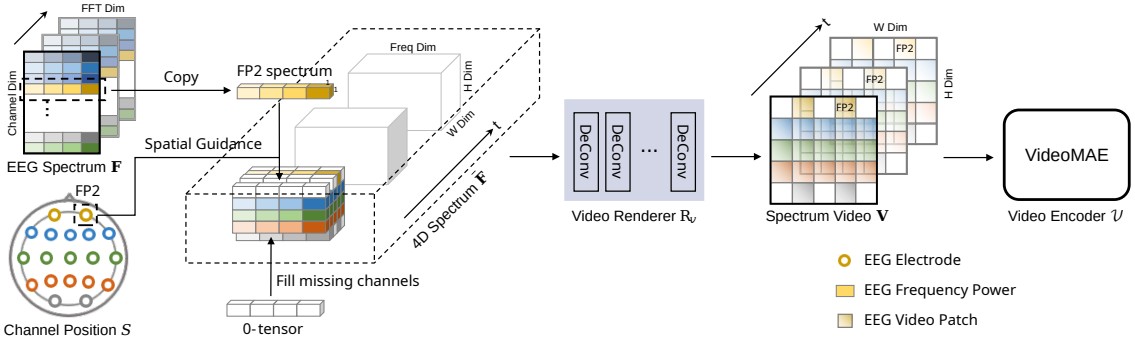

*Figure 2.* Overview of the proposed framework. (i) A renderer $\mathcal{R}_v$ spatializes EEG signals $\mathbf{X}$ into a spatiotemporal spectrum video $\mathbf{V}$; (ii) we utilize a video encoder $\mathcal{V}$ to capture geometry-aware spatiotemporal dynamics from $\mathbf{V}$.

ing model performance on target datasets with entirely distinct electrode layouts. By isolating performance on seen versus novel, unseen channels, this approach provides a comprehensive verification of the model's ability to generalize to various EEG montage settings. (3) We establish **subject-level few-shot learning** as a rigorous evaluation benchmark for EEG foundation models. Inspired by cross-subject transfer learning (Li et al., 2026), this paradigm treats each subject as a distinct task and evaluates rapid adaptation from only a few calibration sessions, directly reflecting real-world EEG deployment scenarios. (4) Through extensive experiments across multiple datasets and tasks, we demonstrate that BSR consistently outperforms prior EEG representation learning methods, establishing a scalable, interpretable, and data-efficient foundation for generalizable EEG modeling.

## 2. Related Works

**Few-shot Learning Paradigms for EEG Foundation Models.** Benchmarking EEG foundation models is an evolving field of research, with several few-shot learning paradigms recently proposed. AdaBrain-Bench (Wu et al., 2025), for instance, introduces a few-shot evaluation protocol that utilizes a fixed proportion of the training data. Similarly, BrainWave (Yuan et al., 2024) explores few-shot learning by evaluating models under 3-shot and 8-shot settings across tasks. In contrast to these approaches, which evaluate few-shot learning under fixed data splits or within-dataset settings, we establish *subject-level few-shot learning* as a benchmark for assessing EEG foundation models. Building on the cross-subject transfer learning protocol (Li et al., 2026), our paradigm requires pretrained models to rapidly adapt to entirely new subjects using only a few calibration sessions, rigorously probing their generalization capabilities under realistic deployment conditions.

**VideoMAE for Few-shot Learning.** VideoMAE (Tong et al., 2022) represents a breakthrough in self-supervised video representation learning, leveraging large-scale unla-

beled video data to learn powerful, generalizable features that excel in few-shot settings. Its masked autoencoding paradigm enables the model to capture rich spatiotemporal dependencies efficiently, making it highly robust for cross-domain generalization. For example, Hatano et al. (2024) show that VideoMAE achieves significant gains in cross-domain few-shot action recognition by training separate models on multiple modalities and optimizing for domain-invariant features. Samarasinghe et al. (2023) demonstrate that a VideoMAE-pretrained universal encoder can transfer effectively to unseen domains in few-shot video understanding tasks. We adopt VideoMAE as our backbone because its design naturally aligns with Brain Signal Rendering (BSR), which converts EEG into spatiotemporal "video" sequences encoding spectral and spatial neural dynamics. VideoMAE's strength in capturing rich spatiotemporal patterns and its efficiency in low-data regimes make it ideal for spectrum videos.

**Alternative EEG-to-Video Approaches.** While the direct transformation of spectral information into video representations was introduced by Bashivan et al. (2016), scaling this method for large-scale pre-training presents substantial computational challenges with current rendering APIs. Besides, Yang et al. (2021b) provides another interesting EEG-to-Visual approach and shows competitive performance in motor imagery decoding. Nevertheless, we remain highly interested in the potential of such pipelines when combined with video foundation models and are actively developing a high-parallelization implementation to facilitate future investigations.

## 3. Methods

**Task Definition.** Let a raw EEG sample be represented as $\mathbf{X} \in \mathbb{R}^{c \times l}$, where $c$ is the number of channels and $l$ is the sequence length. Our goal is to learn a mapping from $\mathbf{X}$ to a task-specific one-hot label vector $\boldsymbol{y} \in \{0, 1\}^m$, where $m$ denotes the number of classes.

**Framework overview.** Our framework consists of three

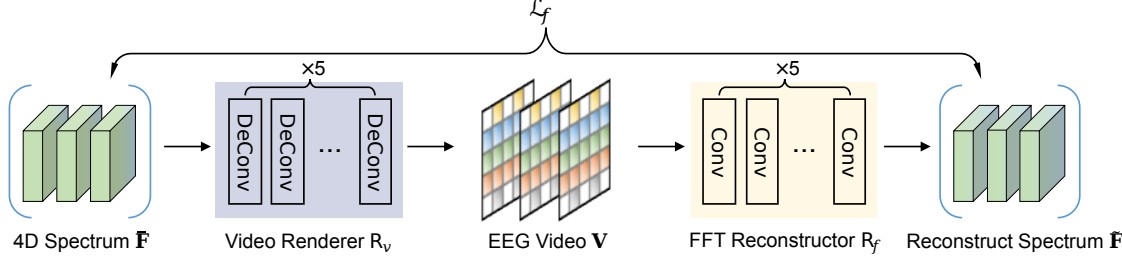

*Figure 3.* Schematic of the BSR Render-Reconstruct pipeline. The renderer $\mathcal{R}_v$ is self-supervisedly pre-trained by reconstructing the original 4D spatiotemporal spectra from the rendered spectrum videos.

stages as in Fig. 2: (i) A Renderer $\mathcal{R}_v$ transforms the EEG signal $\boldsymbol{X}$ into a spatial-temporal spectrum video $\boldsymbol{V} \in \mathbb{R}^{H \times W \times 3}$; (ii) A video encoder $\mathcal{V}$ is pretrained to capture spatiotemporal features from $\boldsymbol{V}$; (iii) The encoder $\mathcal{V}$ is fine-tuned on downstream EEG tasks to produce the final prediction $\boldsymbol{y}$.

### 3.1. EEG to Spectrum Video Transformation

**Spatial-Temporal Spectrum Preprocessing.** Firstly, we preprocess the raw EEG data separately along its temporal and spatial dimensions, as in Fig. 3. To capture the time-varying frequency content of EEG signals, we apply the Short-Time Fourier Transform (STFT), denoted as $\mathrm{stft}(\cdot)$, which decomposes the signal into a sequence of frequency spectrum over time. This operation extracts both frequency and amplitude information from the inherently non-stationary EEG data, thereby enhancing its temporal-frequency representation. We then compute the magnitude ($\mathrm{abs}(\cdot)$) of the resulting complex spectra to obtain the final temporal feature map $\mathbf{F}$:

$$\mathbf{F} = \mathrm{abs}\left(\mathrm{stft}(\boldsymbol{X}; t, d)\right), \quad \mathbf{F} \in \mathbb{R}^{n \times c \times f}, \tag{1}$$

where $t$ and $d$ denote the STFT window size and hop length, respectively; $n = 1 + \lfloor \frac{l-t}{d} \rfloor$ is the number of time windows, $c$ is the number of EEG channels, and $f$ is the number of frequency bins.

While $\mathbf{F}$ contains rich time–frequency information, it lacks explicit spatial encoding. Since each channel in $\mathbf{F}$ corresponds to an EEG electrode with known spatial coordinates on the scalp, we exploit this inherent spatial structure to embed positional information through channel rearrangement. To formalize this process, we define a user-specified channel spatialization map matrix $\boldsymbol{S} \in \mathbb{N}^{h \times w}$, where each element specifies the target spatial location for the corresponding channel. For instance, if $\boldsymbol{S}[1, 4] = 4$ and $\mathbf{F}[:, 4]$ corresponds to electrode FP2, this indicates that the FP2 spectrum should be rendered at the fourth patch in the first row of the spatial map, as illustrated in Figure 2. We then apply a spectrum spatialization algorithm to transform $\mathbf{F}$ into a spatially organized representation:

$$\overline{\mathbf{F}} = \mathcal{S}(\mathbf{F}, \boldsymbol{S}), \quad \overline{\mathbf{F}} \in \mathbb{R}^{n \times h \times w \times f}, \tag{2}$$

where $\mathcal{S}(\cdot)$ denotes the spatialization operation, and $\overline{\mathbf{F}}$ is a structured spatiotemporal EEG feature map suitable for subsequent multimodal processing.

---

**Algorithm 1** Spectrum Spatialization $\mathcal{S}$

---

**Input:** fourier amplitude spectrum $\mathbf{F} \in \mathbb{R}^{n \times c \times f}$, spatialization map $\boldsymbol{S} \in \mathbb{N}^{h \times w}$
Initialize spatialized 4D spectrum tensor $\overline{\mathbf{F}}$ of size $n \times h \times w \times f$.
**for** $i = 1$ to $h$ **do**
    **for** $j = 1$ to $w$ **do**
        **if** $\boldsymbol{S}[i, j] > 0$ **then**
            $\overline{\mathbf{F}}[:, i, j] \leftarrow \mathbf{F}\big[:, \boldsymbol{S}[i, j]\big]$
        **else**
            $\overline{\mathbf{F}}[:, i, j] \leftarrow \mathbf{0}^{t \times n}$
        **end if**
    **end for**
**end for**
**Return** $\overline{\mathbf{F}}$

---

**Rendering Process.** After obtaining the 4D Fourier frequency map $\overline{\mathbf{F}} \in \mathbb{R}^{n \times h \times w \times f}$, we transform it into a structured EEG video representation $\mathbf{V} \in \mathbb{R}^{n \times H \times W \times 3}$ using our *spectrum renderer* $\mathrm{R}_v$, as formalized in Equation (3). The renderer $\mathrm{R}_v$ comprises a sequence of cascaded deconvolution (transposed convolution) layers with equal kernel size and stride, which preserves the time–frequency content while mapping the spatialized EEG features into a dense RGB representation. Specifically, given the spatialized spectrum corresponding to a single time window $\overline{\mathbf{F}}_i \in \mathbb{R}^{h \times w \times f}$, the renderer produces an image-shaped tensor $\mathbf{V}_i \in \mathbb{R}^{H \times W \times 3}$:

$$\mathbf{V} = \left[\mathbf{V}_1, \mathbf{V}_2, \ldots, \mathbf{V}_n\right]^{\top}, \quad \mathbf{V}_i = \mathrm{R}_v(\overline{\mathbf{F}}_i), \tag{3}$$

where $H$ and $W$ are hyperparameters of the renderer defining the spatial resolution of each frame, and $n$ denotes the number of time windows. The resulting tensor $\mathbf{V}$ constitutes a spatiotemporal sequence, effectively an *Spectrum video*, which preserves both spectral and spatial information for downstream video-based processing.

**Reconstruction Process.** To train the renderer $\mathrm{R}_v$, we jointly learn a *reconstructor* $\mathrm{R}_f$ that inverts the rendering

process by reconstructing the spatialized Spectrum $\overline{\mathbf{F}}_i$, as formalized in Figure 3. The reconstructor produces $\tilde{\mathbf{F}}_i \in \mathbb{R}^{h \times w \times f}$, as formalized in Equation (4):

$$\tilde{\mathbf{F}} = [\tilde{\mathbf{F}}_1, \tilde{\mathbf{F}}_2, \ldots, \tilde{\mathbf{F}}_n]^\top, \quad \tilde{\mathbf{F}}_i = \mathrm{R}_f(\mathbf{V}_i), \qquad (4)$$

where $\tilde{\mathbf{F}} \in \mathbb{R}^{n \times h \times w \times f}$ is the reconstructed spatialized feature map. The reconstructor $\mathrm{R}_f$ adopts a symmetrical architecture to the renderer, replacing each deconvolution (transposed convolution) layer with a corresponding convolution layer, while maintaining identical kernel dimensions, stride, and layer depth. This symmetry ensures effective inversion of the rendering process while preserving spectral and spatial information. Implementations of the renderer and reconstructor used in this study are detailed in Table 4.

The entire system is trained end-to-end with an L1 reconstruction loss, defined as:

$$\mathcal{L}_f = \frac{1}{N} \sum_{i=1}^{N} \left| \overline{\mathbf{F}} - \tilde{\mathbf{F}} \right|, \quad N = n \times h \times w \times f, \quad (5)$$

where $N$ is the total number of elements in $\overline{\mathbf{F}}$ and $\tilde{\mathbf{F}}$, ensuring the loss measures the element-wise absolute error over the entire spatiotemporal frequency representation.

The rendered output $\mathbf{V} \in \mathbb{R}^{n \times H \times W \times 3}$ possesses the same spatiotemporal properties as ordinary video inputs. Since no numerical range constraints are imposed during the rendering stage, we normalize each frame $\mathbf{V}_i$ using Contrast Limited Adaptive Histogram Equalization (CL-AHE), denoted as $\mathrm{T}(\cdot)$, to obtain the final video representation:

$$\hat{\mathbf{V}}_i = \mathrm{T}(\mathbf{V}_i), \quad i = 1, 2, \ldots, n. \qquad (6)$$

This normalization ensures consistent intensity distribution across frames, enhancing the stability and performance of subsequent video-based processing.

## 3.2. Pre-Training Video Encoder

In this work, we adopt **VideoMAE** (Tong et al., 2022) as our pre-trained video encoder $\mathcal{V}$, motivated by its strong capability to capture spatiotemporal patterns through masked autoencoding and its superior generalization performance. The VideoMAE pipeline typically involves: (i) partitioning the input video into patches of size $2 \times 16 \times 16$ (representing temporal, height, and width dimensions, respectively); and (ii) employing a ViT backbone with joint space-time attention to process these patches. Before being fed into VideoMAE, all rendered EEG videos $v$ are resized to a resolution of $(224, 224)$ and temporally sampled to 16 frames, ensuring compatibility with the pre-trained encoder and enabling efficient fine-tuning.

**Self-Supervised Pre-training.** We follow the pipeline of Tong et al. (2022) to pretrain VideoMAE. A reconstruction

head is added after the video encoder to form reconstruction video model $\hat{\mathcal{V}}$, which output the reconstructed video $\tilde{\mathbf{V}}$ with same shape of $\mathbf{V}$:

$$\tilde{\mathbf{V}} = \hat{\mathcal{V}}(\mathbf{V}, \boldsymbol{b}), \qquad (7)$$

where $\boldsymbol{b}$ is a boolean vector where 1 denotes masking and 0 denotes not masking.

MSE loss is calculated only between masked patches $\Omega$ and their reconstructed ones:

$$\mathcal{L}_f = \frac{1}{\sum_i \boldsymbol{b}_i} \sum_{i; \boldsymbol{b}_i = 1} |\mathbf{V}(i) - \tilde{\mathbf{V}}(i)|^2. \qquad (8)$$

## 3.3. Subject-level Few-shot Learning

A key challenge in EEG analysis is the substantial variability across subjects, arising from differences in acquisition equipment, sampling protocols, and individual neurophysiological characteristics. This inter-subject variability often leads to poor generalization of models trained on existing datasets when applied to new individuals, thereby limiting the practical applicability of EEG-based systems in real-world scenarios.

The motivation for *subject-level few-shot learning* is to explicitly evaluate and improve a model's ability to adapt to new subjects using minimal labeled data. This setting reflects realistic application scenarios, such as personalized brain-computer interfaces, where collecting extensive labeled EEG data for every new user with new settings is impractical.

To this end, we establish a benchmark task called *subject-level few-shot learning*. Building on the cross-subject transfer learning protocol (Li et al., 2026), we treat each subject as a distinct task with per-subject calibration. For a new subject $s$, we treat all sampled data from that subject as the subject-specific dataset $\mathcal{D}_s = \{(\boldsymbol{X}_{s,j}, \boldsymbol{y}_{s,j})\}_{j=1}^{N_s}$, where $N_s$ denotes the total number of samples available. We divide $\mathcal{D}_s$ into a small training subset $\mathcal{D}_s^{\text{train}}$ and a testing subset $\mathcal{D}_s^{\text{test}}$, with $|\mathcal{D}_s^{\text{train}}| \ll |\mathcal{D}_s|$.

The objective is to fine-tune the pre-trained VideoMAE model using only $\mathcal{D}_s^{\text{train}}$, and then evaluate its performance on $\mathcal{D}_s^{\text{test}}$:

$$\min_{\theta_{\mathcal{V}}, \theta_{\mathcal{H}_s}} \mathcal{L}^{(s)}\big(\mathcal{H}_s \circ \mathcal{V}(\mathcal{R}(\mathcal{D}_s^{\text{train}})), \boldsymbol{y}_s\big), \qquad (9)$$

where $\mathcal{L}^{(s)}$ denotes the loss function for subject $s$ (e.g., cross-entropy), and $\mathcal{R}$ denotes the EEG Video rendering process.

By focusing on rapid adaptation to unseen subjects with only a few samples, *subject-level few-shot learning* provides a realistic and rigorous measure of the generalization ability of EEG video models, and demonstrates the practical advantage of our rendering-based EEG-to-video framework combined with VideoMAE fine-tuning.

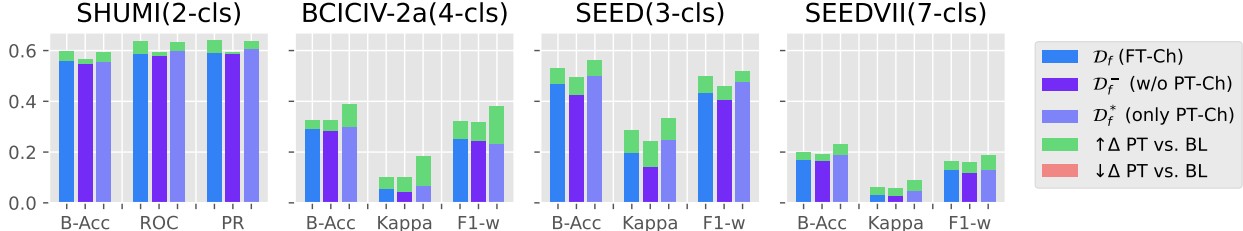

*Figure 4.* Evaluation under the cross-montage fine-tuning and subject-level few-shot learning paradigms. The montage groups represent three distinct generalization scenarios: FT-Ch indicates the full channel set of the fine-tuning dataset, Only PT-Ch denotes channels seen during pre-training, and w/o PT-Ch highlights performance on entirely novel, unseen channels.

### 3.4. Transferring to Unseen Electrode Settings

Another key challenge for general BCI systems is the generalization to different electrode settings. To assess this, we propose the *cross-montage fine-tuning* paradigm. The core strategy involves pre-training the model on large-scale datasets with restricted channel counts, followed by fine-tuning on datasets with distinct electrode layouts. During the finetune, we not only compare the model performance before and after pre-training using all electrodes of the fine-tune dataset, but also repeat this comparison on channels that only exist in pretrained channels and channels not appeared in the pretraining set.

To formulate this paradigm, we first define the electrode selection function $f(\boldsymbol{X}, \mathcal{E})$ that extracts a sub-matrix from $\boldsymbol{X}$ based on the intersection of its original electrode set $\mathcal{E}$ and a target set $\mathcal{E}'$:

$$f(\boldsymbol{X}, \mathcal{E}') = X[\mathcal{E} \cap \mathcal{E}']. \tag{10}$$

Let $\mathcal{D}_p$ and $\mathcal{D}_f$ be the pre-trained and fine-tuned dataset, and $\mathcal{E}_p$ and $\mathcal{E}_f$ be their corresponding channel sets. After pretraining with $\mathcal{D}_p$, we construct 2 additional datasets: (i) only channels that appeared in $\mathcal{D}_p$, denoted by

$$\mathcal{D}_f^* = \{f(\boldsymbol{X}, \mathcal{E}_p), \boldsymbol{X} \in \mathcal{D}_f\}, \tag{11}$$

and (ii) only channels that did not appear in $\mathcal{D}_p$, denoted by

$$\mathcal{D}_f^- = \{f(\boldsymbol{X}, \mathcal{E}_f - \mathcal{E}_p), \boldsymbol{X} \in \mathcal{D}_f\}. \tag{12}$$

We systematically evaluate the model's cross-channel generalization by comparing performance before and after pre-training across $\mathcal{D}_f$, $\mathcal{D}_f^*$, and $\mathcal{D}_f^-$. Specifically: (i) results on $\mathcal{D}_f$ capture the aggregate generalization gain; (ii) $\mathcal{D}_f^*$ reflects improvements derived from refined signal representations and inherent spatiotemporal dynamics; and (iii) $\mathcal{D}_f^-$ isolates the model's ability to learn layout-invariant representations for novel channel configurations.

## 4. Experiments

### 4.1. Datasets

**Spectrum Renderer Pretraining.** TUEG (Obeid & Picone, 2016) is used to pre-train Spectrum Renderer, of 26,846 clinical EEG recordings collected from 2002 to 2017.

**VideoMAE Pre-Training.** Two different pre-training datasets are constructed for different experiments. For *Cross-Montage Fine-Tuning* and experiments on comparing pre-training paradigms, initialization and scaling, we use **TUAB** (Obeid & Picone, 2016) and **TUEV** (Obeid & Picone, 2016) which contain less electrode channels but much more data size comparing with other datasets. The **TUAB** dataset (Obeid & Picone, 2016) is designed for abnormal detection and consists of two categories: normal and abnormal. The **TUEV** dataset (Obeid & Picone, 2016) is an event classification benchmark with six categories, namely spike and sharp wave, generalized periodic epileptiform discharges, periodic lateralized epileptiform discharges, eye movement, artifact, and background. To benchmark against existing methods, we incorporate five additional datasets for self-supervised pre-training: **SEED-DV** (Liu et al., 2024), **Dreamer** (Katsigiannis & Ramzan, 2017), **Physio-MI** (Schalk et al., 2004), **Physio-ERP** (Citi et al., 2010), and **MentalArithmetic** (Zyma et al., 2019) to demonstrate the model's performance under heterogeneous mixed-dataset pre-training scenarios.

**Downstream BCI datasets for Subject-Level Few-Shot Learning.** For subject-level few-shot learning, we use four EEG datasets. The **SEED** dataset (Duan et al., 2013; Zheng & Lu, 2015) targets emotion classification with three categories (negative, neutral, positive), and each subject has three sessions, split into train:validation:test of 1:1:1. The **SEED-VII** dataset (Jiang et al., 2025) extends this to seven emotion categories (happy, surprise, neutral, sad, disgust, fear, anger); each subject has four sessions, but as no single session covers all categories, we used a 2:2 train:test split (sessions 1 and 3 for training, sessions 2 and 4 for testing). The **SHU-MI** dataset (Ma et al., 2022) is a large-scale motor imagery dataset with two classes (left-hand, right-hand), also split 1:1:1 across three sessions. Finally,

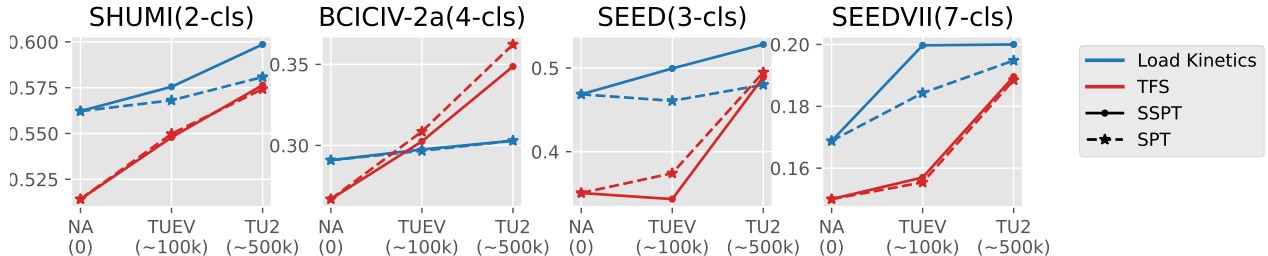

*Figure 5.* Few-shot learning experiment on pre-training paradigms, initialization and scaling. The results compare not only SSPT (Self-Supervised Pre-Training) versus SPT (Supervised Pre-Training), but also model initialization via Load Kinetics (further pre-trained from Kinetics-400 weights) versus those Trained From Scratch (TFS). The X-axis indicates the pre-training dataset used (approximate number of samples), where TU2 denotes the combined TUAB and TUEV dataset.

the well-known **BCICIV-2a** dataset ([Brunner et al., 2008](#)) focuses on motor imagery with four classes (left-hand, right-hand, both feet, tongue), where we adopt the official 1:1 train:test split.

**Downstream BCI datasets for Full-Dataset Fine-Tuning.** We use **SEED-V** and **SHU-MI** for full-dataset fine-tuning evaluation. For **SHU-MI**, we setup a train:val:test split of 3:1:1 with random order. **SEED-V** adopts similar paradigm like **SEED** and **SEED-VII**, which contains 5 emotion categories. We use the subject 1-10 to train, 11-13 to validation and 14-16 to test.

Detailed dataset descriptions encompassing preprocessing steps are deferred to Appendix A.

### 4.2. Experimental Setup

**Evaluation Metrics.** We evaluate model performance using a set of metrics tailored to each task. For **binary classification**, we report balanced accuracy (**B-Acc.**), area under the receiver operating characteristic curve (**AUROC**), and area under the precision-recall curve (**AU-PR**), where AU-PR is particularly robust for imbalanced datasets by focusing on the positive class. For **multi-class classification**, we report balanced accuracy, **Cohen's Kappa** ($\kappa$), which adjusts for chance agreement between predictions and labels, and the **weighted F1 score (F1w)**, the harmonic mean of precision and recall weighted by class sample sizes. For subject-level few-shot learning, we report the aggregate average across all subjects. For full-dataset fine-tuning, results are reported as mean $\pm$ standard deviation over three independent runs with 3 distinct random seeds.

**Experiment Platform.** All experiments were conducted on a machine with $8 \times 80G$ GPUs, an Intel Xeon Gold 6330 CPU, and 200 GB RAM, using Python3.11.11, PyTorch2.5.1, and CUDA12.2. Video I/O was implemented with OpenCV-Python and PIL.

We refer to the VideoMAE model integrated with the BSR framework as **BSR-VideoMAE** throughout these experiments. For method hyperparameters of BSR-VideoMAE

and baselines are deferred to Appendix B.

### 4.3. Cross-Montage Fine-Tuning

We combine the *cross-montage fine-tuning* with *subject-level few-shot learning*. The visualization detailing these three channel strategies is provided in Appendix E. Under each defined channel strategy, the performance is assessed using two models: the Baseline model, which is the **Kinetics-400** ([Kay et al., 2017](#)) initialized **VideoMAE**; and the pre-trained model, derived from the baseline after further self-supervised pre-training on the **TUAB** and **TUEV** datasets ($\mathcal{D}_p$).

The results of this *cross-montage fine-tuning* are detailed in Figure 4. Generally, the Pre-trained (PT) models, which leveraged self-supervised pre-training, consistently demonstrated performance improvement across all experimental settings compared to the Baseline (BT) models, including the $\mathcal{D}_f^-$ (without any pre-training channels) group. This shows that the BSR framework is not just remembering channel arrangements but learns extendable spatio-temporal dynamics. $\mathcal{D}_f^*$ (Only Pre-training Channels) group generally performs best after pre-training, which shows that the representation learned from pretraining is robust and effective. For easier tasks such as SHU-MI and SEED, $\mathcal{D}_f$ outperforms $\mathcal{D}_f^-$ and performs similar to $\mathcal{D}_f^*$, but for difficult tasks $\mathcal{D}_f$ performs similar with $\mathcal{D}_f^-$. This shows that easier tasks rely more on signal representation robustness, but harder tasks relies more on representation of spatiotemporal dynamics.

### 4.4. Pre-training Paradigms, Initialization and Scaling

To comprehensively evaluate the compatibility of VideoMAE with the BSR framework, we conducted experiments examining the effects of different pretraining paradigms, model initialization strategies, and data volume. The experimental settings were as follows: (i) for pretraining paradigms, we considered both self-supervised pre-training following [Tong et al. (2022)](#) and supervised pre-training; (ii) for model initialization, we compared initialization with

*Table 1.* Subject-level few-shot learning results.

| Method | SEED | | | SEED-VII | | | SHU-MI | | | BCIC-IV-2a | | |
|---|---|---|---|---|---|---|---|---|---|---|---|---|
| | B-Acc. | $\kappa$ | F1w | B-Acc. | $\kappa$ | F1w | B-Acc. | AUROC | AU-PR | B-Acc. | $\kappa$ | F1w |
| FFCL | 0.4100 | 0.1103 | 0.3570 | 0.1682 | 0.0280 | 0.1216 | 0.5271 | 0.5608 | 0.5644 | 0.2935 | 0.0581 | 0.2673 |
| ContraWR | 0.3589 | 0.0353 | 0.2124 | 0.1569 | 0.0237 | 0.0976 | 0.5105 | 0.5886 | 0.5962 | 0.2843 | 0.0458 | 0.2505 |
| C-Trans | 0.4421 | 0.1594 | 0.3322 | 0.1629 | 0.0204 | 0.0938 | 0.5388 | 0.5988 | 0.5992 | 0.2531 | 0.0041 | 0.1132 |
| BIOT | 0.3677 | 0.0511 | 0.2245 | 0.1928 | 0.0524 | 0.1303 | 0.5456 | 0.5790 | 0.5828 | 0.2735 | 0.0314 | 0.1931 |
| LaBraM | 0.5263 | 0.2768 | 0.4840 | 0.1554 | 0.0115 | 0.0740 | 0.5415 | 0.5417 | 0.5524 | 0.2836 | 0.0448 | 0.2330 |
| Ours | **0.5815** | **0.3650** | **0.5461** | **0.2054** | **0.0670** | **0.1696** | **0.5959** | **0.6282** | **0.6327** | **0.3275** | **0.1034** | **0.3114** |

*Table 2.* Full-dataset fine-tuning results.

| Method | SEED-V | | | SHU-MI | | |
|---|---|---|---|---|---|---|
| | B-Acc. | $\kappa$ | F1w | B-Acc. | AUROC | AU-PR |
| FFCL | $0.2378 \pm 0.0042$ | $0.0425 \pm 0.0054$ | $0.2202 \pm 0.0159$ | $0.6194 \pm 0.0296$ | $0.7250 \pm 0.0143$ | $0.7125 \pm 0.0275$ |
| ContraWR | $0.2574 \pm 0.0134$ | $0.0643 \pm 0.0178$ | $0.1923 \pm 0.0497$ | $0.6008 \pm 0.0052$ | $0.7224 \pm 0.0051$ | $0.7333 \pm 0.0027$ |
| C-Trans | $0.2612 \pm 0.0232$ | $0.0717 \pm 0.0277$ | $0.2026 \pm 0.0440$ | $0.6160 \pm 0.0278$ | $0.6742 \pm 0.0285$ | $0.6879 \pm 0.0340$ |
| BIOT | $0.2719 \pm 0.0142$ | $0.0937 \pm 0.0154$ | $0.2679 \pm 0.0113$ | $0.5614 \pm 0.0332$ | $0.5839 \pm 0.0480$ | $0.5797 \pm 0.0469$ |
| LaBraM | $0.2411 \pm 0.0030$ | $0.0514 \pm 0.0068$ | $0.2354 \pm 0.0096$ | $0.6421 \pm 0.0128$ | $0.6938 \pm 0.0325$ | $0.6737 \pm 0.0627$ |
| Ours | $\mathbf{0.3155 \pm 0.0127}$ | $\mathbf{0.1444 \pm 0.0159}$ | $\mathbf{0.3160 \pm 0.0186}$ | $\mathbf{0.6608 \pm 0.0068}$ | $\mathbf{0.7324 \pm 0.0014}$ | $\mathbf{0.7396 \pm 0.0032}$ |

Kinetics-400 (Kay et al., 2017) weights against training from scratch; (iii) for data scaling, we compared no pretraining **(NA)**, pretraining only on **TUEV**, and pretraining on both TUAB and TUEV **(TU2)**.

As shown in Figure 5, the scaling experiment reveals three key observations. First, both self-supervised and supervised pretraining prove effective, with self-supervised pretraining generally yielding slightly better performance. Second, initializing from Kinetics weights not only provides a favorable starting point but also leads to better downstream performance in most cases under comparable pretraining resources. Finally, model performance improves consistently with increased volume of EEG pretraining data, confirming the expected positive relationship between domain-specific data scale and model enhancement.

### 4.5. Quantitative Results

**Baselines.** To evaluate the performance of the proposed BSR-VideoMAE with state-of-the-art EEG models, we compared against five FFT-based baselines. **FFCL** (Li et al., 2022) uses a CNN-LSTM fusion network for motor imagery classification, combining spatial and temporal features. **ContraWR** (Yang et al., 2021a) applies self-supervised learning to improve sleep staging by leveraging unlabeled EEG data. **C-Trans** (Peh et al., 2022) employs a CNN-Transformer hybrid with belief matching loss for multi-type EEG artifact detection, maximizing artifact rejection while preserving clean signals. **BIOT** (Yang et al., 2023a) presents a flexible biosignal encoder for multi-dataset pre-training and task-specific fine-tuning across diverse EEG formats.

**LaBraM** (Jiang et al., 2024) proposes a unified EEG foundation model to address the limitations of specialized deep learning approaches. For **LaBraM**, we load the official pretrained weights, while other methods are trained from scratch.

We extend the pre-train dataset size to **TUAB**, **TUEV**, **SEED-DV**, **Dreamer**, **PhysioMI**, **PhysioERP** and **MentalArithmetic** and trained the VideoMAE-base for 50 epochs. Both subject-level few-shot learning and full-dataset fine-tuning are tested, and corresponding results are shown in Table 1 and Table 2.

Across all experimental regimes, BSR-VideoMAE consistently establishes a new performance benchmark, demonstrating robust superiority over existing Spectrum-based methods. In the challenging subject-level few-shot learning scenarios, where models must generalize to unseen subjects with minimal calibration, our framework delivers substantial performance gains. On the SHU-MI dataset, BSR-VideoMAE achieves a Balanced Accuracy of 59.59%, outperforming the previous best-performing method by a significant margin of approximately 5.0%. On the SEED dataset, the improvement is even more pronounced: our model secures a Balanced Accuracy of 58.15%, representing an absolute increase of over 5%.

These few-shot advantages are further corroborated by full-dataset fine-tuning results. notably on SEED-V, where BSR-VideoMAE achieves a Balanced Accuracy of 31.55%, surpassing the closest state-of-the-art competitor by 4.36%. Collectively, these quantitative results confirm that BSR-VideoMAE delivers superior generalization capabilities. It

consistently breaks the performance ceilings of traditional channel-dependent approaches, ensuring stability across varying datasets and acquisition protocols.

## 5. Conclusions and Future Works

We present **Brain Signal Rendering (BSR)**, a framework that spatializes EEG signals into structured spectrum video representations to overcome the long-standing challenges of sensor heterogeneity and rigid architectural biases. By leveraging the spatiotemporal inductive biases of video foundation models, BSR enables effective knowledge transfer across diverse tasks and electrode configurations. Extensive evaluations in *subject-level few-shot* and *cross-montage fine-tuning* settings confirm that BSR-VideoMAE significantly outperforms existing spectrum-based models, establishing a new state-of-the-art for generalizable and data-efficient neural decoding.

Looking ahead, two promising directions can further amplify this impact: (1) *Efficiency scaling*—exploring lighter or EEG-specialized encoders to reduce computational cost without sacrificing performance, and pre-training on larger scale combined EEG datasets; (2) *Rendering exploration*—investigating alternative transformation methods, and using metrics which could reflect brain activity such as band features for rendering evaluation. These avenues promise to advance BSR toward a scalable, adaptable foundation for EEG representation learning, paving the way for practical, high-performance BCI systems.

## Acknowledgment

Prof. Yanwei Fu is with the School of Data Science and MOE Frontiers Center for Brain Science, Shanghai Key Lab of Intelligent Information Processing Fudan University, Shanghai Innovation Institute, and Fudan ISTBI–ZJNU Algorithm Centre for Brain-inspired Intelligence, Zhejiang Normal University.

This work was supported by Brain Science and Brain-like Intelligence Technology - National Science and Technology Major Project (2021ZD0200500), the National Natural Science Foundation of China (62472206, 3254100307), National Key R&D Program of China (2025YFC3410000), Shenzhen Science and Technology Innovation Committee (RCYX20231211090405003, JCYJ20220818100213029), Guangdong Basic and Applied Basic Research Foundation (2026B1515020099), Guangdong S&T Program (Grant No. 2026B0101110003), Shanghai Municipal Special Program for Basic Research on General AI Foundation Models (2025SHZDZX026D05), Shenzhen Loop Area Institute under grant FPF10120250012, and the open research fund of the Guangdong Provincial Key Laboratory of Mathematical and Neural Dynamical Systems, the Center for Computational Science and Engineering at Southern University of Science and Technology, Shenzhen Key Laboratory of Smart Healthcare Engineering.

## Impact Statement

This paper presents work whose goal is to advance the field of machine learning. There are many potential societal consequences of our work, none of which we feel must be specifically highlighted here.

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

## A. Datasets

**Data Pre-processing.**

We preprocess the datasets as follows:

- For TUEG, TUAB and TUEV, we follow the pipeline of (Jiang et al., 2024), which employs a filter between 0.1Hz and 75Hz, as well as a notch filter of 60Hz.

- For SEED, SEED-VII and SEED-DV, we directly use the official preprocessed data which has already been downsampled to 200HZ.

- For SEED-V, we follow the pipeline of (Wang et al., 2025), which applies a band filter from 0.3Hz to 75 Hz.

- For SHUMI, we follow the pipeline of (Wang et al., 2025), which takes no preprocess except the down-sampling.

- For BCICIV-2a dataset, we employ a band filter from 0Hz to 38 Hz.

- For MentalArithmetic dataset, we applied a filter between 0.1Hz and 75Hz, as well as a notch filter of 50Hz.

- For Physio-MI, we filtered signal below 0.3HZ and applied notch filter of 60HZ.

- For Physio-ERP, we followed the pipeline of (Wang et al., 2024), which applied a filter between 0.1Hz and 120Hz.

All EEG data are down-sampled to 200 Hz and stored in **unipolar** form. Dataset channels, segment duration, paradigms and usage are summarized in Table 3. Discrepancy (of the full-dataset fine-tuning tasks) between the results reported in our paper and the results reported in the original papers of baseline methods came from both difference in pre-processing and software/hardware difference. In particular, the length of all data in SEED, SEED-V and SEED-VII datasets are split into **10 seconds** in our study, which might be different from settings in (Wang et al., 2025) (which utilizes 1 second segment).

| Dataset | #Ch | Segment Duration | Paradigm | Usage |
|---|---|---|---|---|
| TUEG | 19 | 10 seconds | – | Pre-training BSR Renderer |
| TUAB | 19 | 10 seconds | Abnormal Detection | Pre-training BSR-VideoMAE |
| TUEV | 19 | 5 seconds | Event Classification | Pre-training BSR-VideoMAE |
| MentalArithmetic | 19 | 4 seconds | Mental Stress Detection | Pre-training BSR-VideoMAE |
| Physio-MI | 64 | 10 seconds | Motor Imagery | Pre-training BSR-VideoMAE |
| Physio-ERP | 64 | 2 seconds | Event-Related Potentials | Pre-training BSR-VideoMAE |
| SEED-DV | 62 | 10 seconds | Visual Perception | Pre-training BSR-VideoMAE |
| SEED-V | 62 | 10 seconds | Emotion Recognition | Fine-tuning |
| SHU-MI | 32 | 4 seconds | Motor Imagery | Fine-tuning and Few-Shot Learning |
| SEED | 62 | 10 seconds | Emotion Recognition | Few-Shot Learning |
| SEED-VII | 62 | 10 seconds | Emotion Recognition | Few-Shot Learning |
| BCIC-IV-2a | 22 | 4 seconds | Motor Imagery | Few-Shot Learning |

*Table 3.* Summarization of dataset paradigms and usage.

To ensure the train-(val)-test split for all methods is strictly consistent, the rendering process on all evaluation datasets does not access the raw data but the processed EEG segments, which are the direct input of all baseline methods.

## B. Implementation Details.

**Hyperparameters for Subject-Level Few-Shot Learning.**

- BSR-VideoMAE will be trained for 25 epochs for all experiments, with a learning rate using the AdamW optimizer with a learning rate of 1e-5, weight decay of 1e-4 and a cosine annealing scheduler.

- LaBraM is trained for 50 epochs with its recommended hyperparameters for all experiments.

- For other baselines, if validation set is applicable, they will be trained for 50 epochs with an early stop callback on AUROC (binary classification) or $\kappa$ (multiclass classification). Elsewhere they will be trained for 15 epochs to prevent overfitting. Other hyperparameters are referred to (Yang et al., 2023a).

**Hyperparameters for Full-Dataset Fine-Tuning.**

- BSR-VideoMAE will be trained for 20 epochs for all experiments, with a learning rate using the AdamW optimizer with a learning rate of 1e-5, weight decay of 1e-5 and a cosine annealing scheduler.

- LaBraM is trained for 50 epochs with its recommended hyperparameters for all experiments.

- For other baselines, they will be trained for 100 epochs with an early stop callback on AUROC (binary classification) or $\kappa$ (multiclass classification). Other hyperparameters are referred to (Yang et al., 2023a).

**BSR-VideoMAE Implementation.** The architectural specifications of the BSR Renderer and Reconstructor are detailed in Table 4. Our implementation of VideoMAE is adapted from the Hugging Face library, specifically utilizing the Base variant (approx. 86M parameters) for all experiments.

*Table 4.* Hyperparameters for BSR Renderer and Reconstructor

| Layer | Renderer - ConvTranspose2d (in-ch, out-ch, kernel-size, stride) | Reconstructor - Conv2d (in-ch, out-ch, kernel-size, stride) |
|---|---|---|
| 1 | (101, 50, 2, 2) | (3, 6, 2, 2) |
| 2 | (50, 25, 2, 2) | (6, 12, 2, 2) |
| 3 | (25, 12, 2, 2) | (12, 25, 2, 2) |
| 4 | (12, 6, 2, 2) | (25, 50, 2, 2) |
| 5 | (6, 3, 2, 2) | (50, 101, 2, 2) |

## C. Additional Ablation Studies

### C.1. Ablation on pretrain initialization

To decouple the contributions of the BSR rendering framework from the pre-training data scale and initialization, we re-evaluated on both few-shot (Table 5) and full-dataset fine-tuning (Table 6) using existing model checkpoints, comparing three variants of BSR-VideoMAE: **TU2-TFS**, trained from scratch on TUAB+TUEV (EEG only); **TU2+Kinetics**, initialized from Kinetics-400 (Kay et al., 2017) then pre-trained on TU2; and **Full**, initialized from Kinetics-400 then pre-trained on all 7 EEG datasets. Note that LaBraM's pre-training includes SEED (see their Appendix D), and therefore its SEED result is marked with * and serves only for reference.

*Table 5.* Few-Shot Learning Results on pretrain checkpoints (B-ACC)

| Method | Pretrain | SEED | SEED-VII | SHU-MI | BCIC-IV-2a |
|---|---|---|---|---|---|
| VideoMAE | TU2-TFS | 0.4890 | 0.1896 | 0.5764 | 0.3487 |
| VideoMAE | TU2+Kinetics | 0.5282 | 0.2000 | 0.5986 | 0.3244 |
| VideoMAE | Full | 0.5815 | 0.2054 | 0.5959 | 0.3275 |
| BIOT | – | 0.3677 | 0.1928 | 0.5456 | 0.2735 |
| LaBraM | Official | 0.5263* | 0.1554 | 0.5415 | 0.2836 |

As shown in Table 5, even the TU2-TFS model, which is trained from scratch on EEG data alone, substantially outperforms all existing baselines across most few-shot datasets. The gains from Kinetics-400 initialization and larger-scale pre-training are incremental in this setting, suggesting that the strong few-shot performance of BSR-VideoMAE is primarily driven by the BSR rendering framework itself.

In contrast, under full-dataset fine-tuning (Table 6), larger-scale pre-training yields clear and substantial gains. For instance, on SEED-V, the Full variant achieves 31.55% B-Acc., compared to 28.03% (TU2+Kinetics) and 26.44% (TU2-TFS), confirming that model capacity benefits significantly from increased pre-training data volume in the full fine-tuning regime.

*Table 6.* Full Fine-tuning Results on pretrain checkpoints (B-ACC)

| Method | Pretrain | SEED-V | SHU-MI |
|---|---|---|---|
| VideoMAE | TU2-TFS | $0.2644 \pm 0.0132$ | $0.5583 \pm 0.0176$ |
| VideoMAE | TU2+Kinetics | $0.2803 \pm 0.0078$ | $0.6182 \pm 0.0271$ |
| VideoMAE | Full | $0.3155 \pm 0.0127$ | $0.6608 \pm 0.0068$ |
| BIOT | – | $0.2719 \pm 0.0142$ | $0.5614 \pm 0.0332$ |
| LaBraM | Official | $0.2411 \pm 0.0030$ | $0.6421 \pm 0.0128$ |

In summary, the state-of-the-art performance of BSR-VideoMAE in few-shot learning is mainly driven by the BSR rendering framework, while in full fine-tuning it mainly benefits from the full pre-training on diverse EEG datasets. Kinetics-400 serves as a useful but non-essential initialization, providing moderate gains that can be compensated by sufficient EEG-specific pre-training.

### C.2. Ablation on bsr-channels

To investigate the effect of the number of rendered channels on EEG representation quality, we conducted an ablation under the full-dataset fine-tuning paradigm with varying numbers of rendered channels $N_c \in \{1, 3, 8\}$. For each setting, we compared training from scratch (TFS) against initialization from Kinetics-400 (Kay et al., 2017) pretrained weights (with modifications to the embed layers to accommodate the channel count). The corresponding Renderer was pretrained on the combined SEED and SHU-MI datasets for 5,000 epochs. Results are summarized in Figure 6.

As shown in Figure 6, the number of rendered channels has a negligible impact on EEG representation quality. Both TFS and Load Kinetics configurations perform similarly across $N_c \in \{1, 3, 8\}$, indicating that BSR does not compress information: the rendered spectrum image patch (size $32 \times 32 \times N_c$) provides a richer representation than the original spectral patch (size $1 \times 1 \times 101$). In contrast, models initialized with Kinetics-400 weights consistently and substantially outperform those trained from scratch across all channel configurations, confirming that data-driven visual priors from natural videos are crucial for effective spectrum video representation learning. We therefore adopt the 3-channel configuration primarily for engineering convenience.

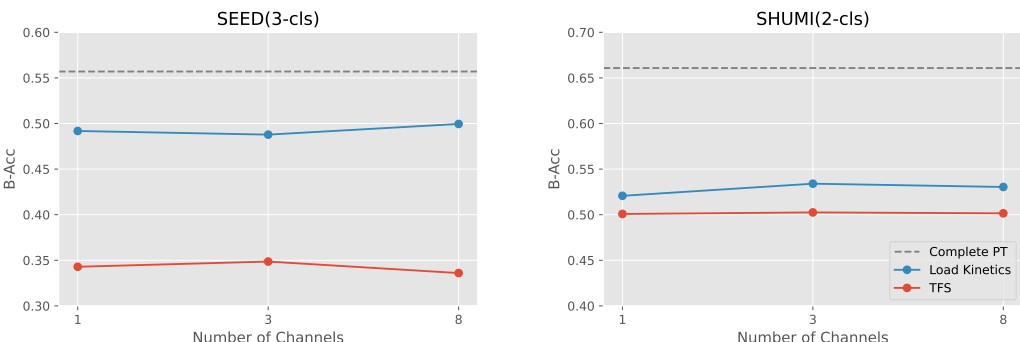

*Figure 6.* Channel Ablation Study. The results compare the performance of the BSR-VideoMAE framework when rendering the spectrum video into different channels. Load Kinetics denotes pretrain from kinetics weights, while TFS denotes train from scratch. The results with complete pre-training with seven datasets (i.e. the model used in table1 and 2) are marked with dashed lines and denoted as complete PT.

## D. Comparing with None-FFT based EEG foundation models

In our initial selection of baselines, we primarily focused on spectrum-based methods to control for the influence of the FFT transform itself (see the discussion on P300 in Appendix F of Wang et al. (2024)), thereby enabling a cleaner validation of the proposed BSR framework. However, to fully demonstrate the superiority of BSR-VideoMAE, it is necessary to compare with recent state-of-the-art models such as **CBraMod** (Wang et al., 2025) and **CSBrain** (Zhou et al., 2026). To this end, we conducted supplementary experiments on full-dataset fine-tuning tasks using the SEED and SHU-MI datasets. The results are presented in Table 7.

As shown in Table 7, BSR-VideoMAE consistently and significantly outperforms CBraMod and CSBrain across all metrics on both datasets. The results demonstrate that the BSR framework, by explicitly modeling electrode spatial geometry and spectral temporal dynamics, learns more generalizable representations than existing non-spectrum EEG foundation models.

*Table 7.* Additional full-dataset fine-tuning results comparing with non-spectrum based SOTAs.

| Method | SEED | | | SHU-MI | | |
| --- | --- | --- | --- | --- | --- | --- |
| | B-Acc. | $\kappa$ | F1w | B-Acc. | AUROC | AU-PR |
| CBraMod | $0.4985 \pm 0.0351$ | $0.2366 \pm 0.0507$ | $0.4540 \pm 0.0436$ | $0.6550 \pm 0.0089$ | $0.7175 \pm 0.0101$ | $0.7092 \pm 0.0137$ |
| CSBrain | $0.4987 \pm 0.0043$ | $0.2374 \pm 0.0076$ | $0.4689 \pm 0.0135$ | $0.6329 \pm 0.0037$ | $0.6983 \pm 0.0050$ | $0.7003 \pm 0.0034$ |
| Ours | $\mathbf{0.5570} \pm 0.0022$ | $\mathbf{0.3239} \pm 0.0027$ | $\mathbf{0.5247} \pm 0.0032$ | $\mathbf{0.6608} \pm 0.0068$ | $\mathbf{0.7324} \pm 0.0014$ | $\mathbf{0.7396} \pm 0.0032$ |

# E. Spectrum Video Visualization

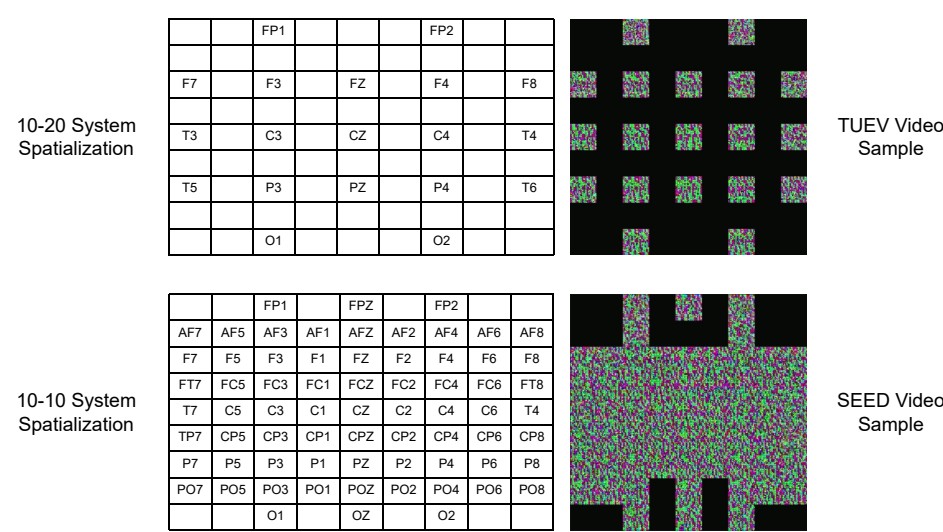

*Figure 7.* Spatialization matrix and EEG video examples.

Figure 7 illustrates the spatialization for both the 10-20 and 10-10 systems, along with video examples from the TUEV and SEED datasets. It is important to note that the BSR renderer's pre-training is independent of any specific EEG system. Once pre-trained, the renderer can be directly applied to any system defined by a user-specified spatialization matrix without requiring further training.

For the three datasets constructed in *cross-montage fine-tuning* experiments, we illustrate the corresponding channel visualization in Fig 8.

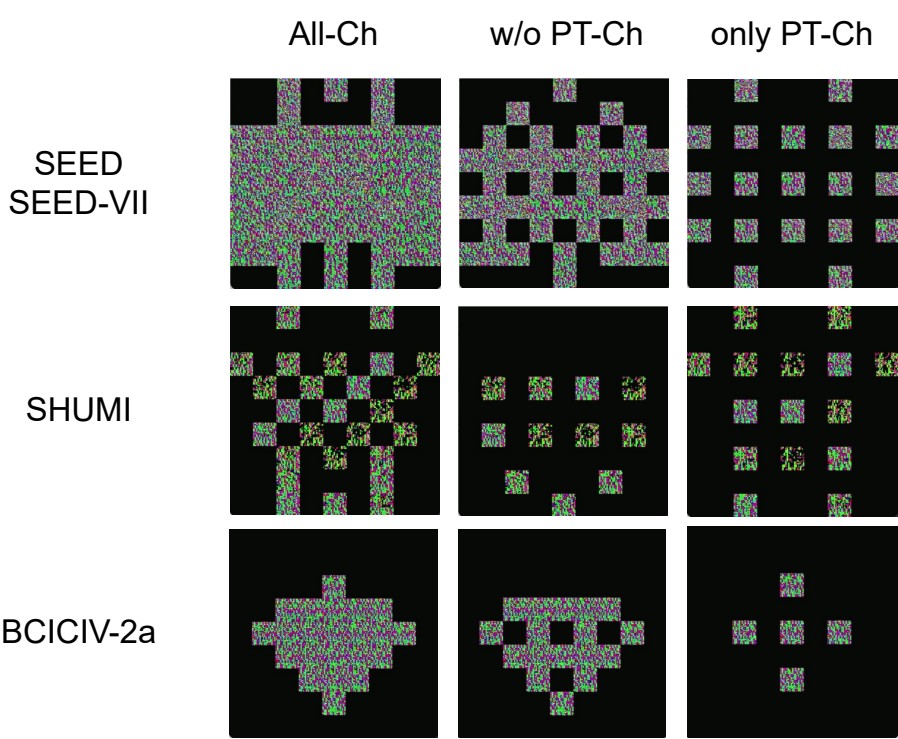

*Figure 8.* Visualization for Cross-Montage Fine-Tuning Experiments.

