# OpenReview forum: "Harnessing Spectrum Video for Subject-Level Few-Shot and Cross-Montage EEG Generalization"
_ICML.cc/2026/Conference — ICML 2026 regular_

### Official Review · Reviewer_nBgg · 2026-03-04

**Soundness:** 2
**Presentation:** 3
**Significance:** 3
**Originality:** 3
**Overall Recommendation:** 4
**Confidence:** 4

**Summary:**

This paper introduces Brain Signal Rendering (BSR), a framework that converts raw EEG signals into "Spectrum Videos" by mapping STFT magnitudes onto a 2D grid based on physical electrode locations. Using a VideoMAE backbone, the model learns spatiotemporal dynamics through self-supervised masked autoencoding. The authors evaluate their approach via cross-montage fine-tuning and subject-level few-shot learning, showing solid performance gains over several recent EEG representation methods.

**Compliance With Llm Reviewing Policy:**

Affirmed.

**Final Justification:**

The authors' response has resolved the issues I raised. Therefore, I would like to increase my score to 4.

**Key Questions For Authors:**

1. How does the 3D-to-2D projection in the BSR framework account for the loss of true geodesic distances between electrodes? Have you analyzed if this spatial distortion negatively impacts the topological features learned by VideoMAE?
2. Regarding the sparsity of the rendered videos (especially for 19-channel setups): What percentage of the VideoMAE patches consist entirely of 0-tensors? How does this sparsity impact the FLOPs, inference latency, and memory footprint compared to native 1D/2D EEG models like LaBraM?
3. Could you provide a table detailing the parameter counts for all baseline models compared to BSR-VideoMAE, along with an analysis of the performance relative to model capacity? Additionally, why were only "FFT-based baselines" selected, and is this scope too narrow to represent the broader landscape of current EEG foundation models?
4. Given the admitted rendering API bottleneck, what is the realistic throughput (e.g., samples processed per second) of the BSR pipeline? How does this framework scale when processing massive, population-level medical datasets (e.g., hundreds of thousands of multi-modal recordings)?

**Limitations:**

No. While the authors briefly mention computational rendering challenges, they overlook several fundamental methodological limitations. A robust limitations section must address: (1) the inherent loss of phase information due to relying solely on STFT magnitudes; (2) the computational inefficiency caused by rendering highly sparse 2D grids; and (3) the geometric distortion introduced by projecting 3D scalp topology onto a 2D matrix.

**Strengths And Weaknesses:**

Strengths:
1. Original Perspective: Reframing electrode heterogeneity as a physical projection problem and leveraging video foundation models (VideoMAE) is a creative and highly original approach to circumvent rigid, channel-first architectural constraints.
2. Rigorous Evaluation Paradigms: Introducing the cross-montage fine-tuning protocol—specifically assessing performance on entirely unseen channels—is a highly valuable contribution that the EEG community currently lacks for benchmarking generalization.

Weaknesses:
1. Spatial Distortion & Topology Loss: The human scalp is a 3D non-Euclidean surface. Projecting electrode coordinates onto a rigid 2D grid matrix ($S∈N^{h×w}$) inherently distorts the true geodesic distances between electrodes. The paper does not address how this spatial distortion impacts the self-attention mechanism, which relies on these spatial relationships to learn neural topology.
2. Inefficiency of Sparse Rendering: As shown in Figure 6, mapping sparse montages (e.g., the 19-channel 10-20 system) to a grid results in a significant number of "empty" regions filled with 0-tensors (Algorithm 1). Since VideoMAE operates on patch embeddings, this highly sparse video representation introduces severe computational redundancy. The authors fail to analyze how these empty patches affect the efficiency of the attention mechanism and overall FLOPs.
3. Loss of Phase Dynamics: By calculating only the magnitude of the STFT to construct the feature map (Eq. 1), the BSR framework completely discards phase information. Phase synchrony is a fundamental biomarker for functional connectivity in EEG. Ignoring phase dynamics limits the model's potential to serve as a comprehensive foundational representation for complex neurophysiological states.
4. Fairness and Scope of Baseline Comparisons: The VideoMAE-base model contains approximately 86M parameters. However, the manuscript does not report the parameter counts of the chosen baselines (FFCL, ContraWR, C-Trans, BIOT, LaBraM) nor does it provide an analysis of the performance trade-off relative to model size. It is crucial to clarify whether the performance gains stem from the BSR representation itself or simply from utilizing a heavier model. Furthermore, the evaluation explicitly states it compares against "five FFT-based baselines" (Section 4.5). The paper lacks a clear justification for exclusively selecting FFT-based methods and excluding other competitive non-FFT or raw-waveform EEG foundation models, raising concerns about whether the baseline selection is too narrow.
5. Scalability Bottlenecks: The authors acknowledge that current rendering APIs present "substantial computational challenges" for large-scale pre-training. As the community moves towards building comprehensive, multi-modal medical foundation models using massive, population-level datasets, this I/O and spatial rendering bottleneck could become a significant limitation. The manuscript lacks a deeper discussion on the practical throughput and scalability limits of this pipeline.

---

> ### Author Rebuttal · Authors · 2026-03-31
>
> **W1 & Q1: Spatial Distortion & Topology Loss**
>
> We believe that while spatial distortion and topology loss do exist in the projection process, they have little to no impact on the VideoMAE model, as the model primarily learns relative spatial relationships. To empirically test whether spatial distortion affects the topological features learned by the pre-trained VideoMAE, we designed an additional rendering pipeline based on the MNE standard-1020 projection. Few-shot learning setting is used to maximize the potential differences in rendering strategies.
>
> Table 1. Few-shot learning results on SEED.
> ||Pretrain|B-ACC|cohen|f1|
> |-|-|-|-|-|
> |VideoMAE 19ch|Kinetics|0.5037|0.2522|0.4759|
> |VideoMAE 19ch topo|Kinetics|0.4977|0.2409|0.4597|
> |VideoMAE|Full|0.5570|0.3239|0.5247|
> |LaBraM|Full|0.5263|0.2768|0.4840|
>
> Table 2. Few-shot learning results on SHU-MI.
> ||Pretrain|B-ACC|roc_auc|pr_auc|
> |-|-|-|-|-|
> |VideoMAE 15ch|Kinetics|0.5558|0.5819|0.5860|
> |VideoMAE 15ch topo|Kinetics|0.5596|0.5863|0.5922|
> |VideoMAE|Full|0.5959|0.6282|0.6327|
> |LaBraM|Full|0.5415|0.5417|0.5524|
>
> The "topo" suffix denotes the rendering pipeline using the MNE 10-20 projection, while 19/15ch refers to using the same channel montage arrangement as in TUAB (i.e. only PT-Ch) to maintain spatial compatibility. The fully pre-trained VideoMAE and LaBraM are included as references. As shown, both rendering pipelines yield nearly identical performance and competitive with LaBraM. This indicates that the spatial distortion introduced by our original grid-based projection has negligible impact on the topological features learned by the Kinetics-pretrained VideoMAE.
>
> **Q2: Sparsity**
>
> Q2.1 Percentage of all‑zero patches
>
> In our default setting, we render EEG onto a 9×9 spatial grid. For a 19‑channel dataset, 77% pixels are rendered by 0-tensor. If we tailored the rendering to 19 channels, a 5×5 grid would reduce the all‑zero patch ratio to 24%.
>
> Q2.2 Impact on FLOPs, latency, and memory
>
> VideoMAE (like standard ViT) does not exploit input sparsity, which means every patch is linearly embedded and participates in self‑attention. Therefore:
>
> - FLOPs are fixed at 361.1253 GFLOP.
> - Inference latency (RTX 4090, serial):
>     - BSR‑VideoMAE (independent of channel count): 17 ms/sample
>     - LaBraM (input length varies with channels):  6 ms/sample for 19 channels, 17 ms/sample for 62 channels.
> - Detailed memory footprint (unit: MB) is calculated as follows:
>
> ||float32|float16|
> |-|-|-|
> |Param|329|164|
> |Grad|329|164|
> |Optim |658|329|
> |Total|1,316|658|
>
> Thus, VideoMAE’s latency is on par with the worst‑case scenario of LaBraM, while achieving substantially better generalization. Although sparsity does not lead to computational savings, the unified video representation enables robust cross‑montage transfer (Figure 4).
>
> **W4 & Q3**
>
> **Performance relative to model capacity**
> The parameters of the models and their performance on SHUMI (using average B-Acc on 3 subjects to save space) are listed as follows:
> |Method|Param|Few-shot|Full fine-tune|
> |--|--|--|--|
> |FFCL| 2.4M | 0.5271 | 0.6194 |
> |ContraWR| 1.6M | 0.5105 | 0.6008 |
> |C-Trans| 3.2M | 0.5388 | 0.6160 |
> |BIOT| 3.2M | 0.5456 | 0.5614 |
> |LaBraM| 5.8M | 0.5415 | 0.6421 |
> |BSR-VideoMAE| 86M | 0.5959 | 0.6608 |
>
> While BSR-VideoMAE possesses a significantly larger parameter count, this increase is not merely for pursuing SOTA performance; more importantly, the spatial redundancy is leveraged to achieve seamless compatibility with various electrode arrays and superior generalization to unseen montages. Furthermore, we will evaluate more VideoMAE variants in future work.
>
> **Why only “FFT-based baselines” are selected**
>
> We appreciate the reviewer's insightful comment regarding the scope of our baselines. Our initial focus on spectrum-based methods was a deliberate design choice to control for the inherent performance boost provided by the FFT transform itself (as discussed in Appendix F of EEGPT [1]). To address the concern regarding the broader landscape of EEG foundation models, we have conducted supplementary experiments comparing with CBraMod and CSBrain on full-dataset fine-tuning (metric: B-Acc):
>
> ||SEED|SHUMI|
> |-|-|-|
> |CBraMod|0.4985|0.6550|
> |CSBrain|0.4987|0.6329|
> |BSR-VideoMAE|0.5570|0.6608|
>
> Supplementary results demonstrate that BSR-VideoMAE consistently achieves superior performance in all metrics.
>
> [1] Wang, Guagnyu, et al. "EEGPT: pretrained transformer for universal and reliable representation of EEG signals." _NeurIPS2024_.
>
> **Q4**
>
> 1. The BSR Renderer is highly efficient, consisting of only five Transposed Convolutional layers with minimal computational overhead. Our benchmarks show that a single RTX4090 can process 100k samples in about 25 minutes.
> 2. Despite the high capacity of the VideoMAE backbone, it maintains a practical throughput of 58 samples/s on RTX4090. By leveraging industry-standard optimizations like TFLite, the throughput further increases to 68 samples/s on a Xeon CPU.

---

> > ### Author Rebuttal · Reviewer_nBgg · 2026-04-03
> >
> > I thank the authors for patiently addressing my questions. I have increased my score.

---

> > > ### Author Response · Authors · 2026-04-03
> > >
> > > Dear Reviewer nBgg,
> > >
> > > We appreciate your thorough and constructive feedback, as well as your positive recognition. Your suggestions are of great help to us in improving this work.
> > >
> > > Best regards, Authors of Paper 20779

---

### Official Review · Reviewer_29UB · 2026-03-05

**Soundness:** 3
**Presentation:** 3
**Significance:** 4
**Originality:** 4
**Overall Recommendation:** 5
**Confidence:** 5

**Summary:**

The present work introduces Brain Signal Rendering (BSR), a framework that maps heterogeneous EEG channels into a shared 2D spatial grid using physical coordinates to create a unified spatiotemporal tensor. Rather than treating EEG channels as flat, independent features, the authors use an autoencoder pipeline (a "renderer" and "reconstructor") to map frequency features (via STFT) into this spatial grid. The resulting spatiotemporal representation is then passed to a pre-trained VideoMAE model, effectively utilizing the learned spatiotemporal priors of a video foundation model to initialize and improve EEG representation learning. The framework is evaluated on two primary settings: cross-subject transfer with calibration and cross-device transfer. Experiments across several benchmark datasets (SEED, SEED-VII, SHU-MI, BCIC-IV-2a) demonstrate that BSR-VideoMAE outperforms existing state-of-the-art EEG foundation models.

**Compliance With Llm Reviewing Policy:**

Affirmed.

**Final Justification:**

The primary strength of this work lies in its originality. The authors introduce a compelling paradigm demonstrating that spatiotemporal priors learned from natural videos can be successfully transferred to an entirely new spatiotemporal domain like EEG. In my initial review, my main concern was understanding whether the state-of-the-art performance was driven by the architecture itself or by the Kinetics-400 prior. The authors provided a thorough rebuttal with the exact ablation studies requested. These results confirmed that the natural video prior is indeed a major driver of the downstream performance. It successfully validates the core premise that visual foundation model priors can be repurposed for structured EEG representation learning. Because the rebuttal transparently resolved my main questions regarding the source of the performance gains, the conceptual originality and strong empirical results clearly outweigh the initial weaknesses. I therefore recommend Accept.

**Key Questions For Authors:**

1. The proposed framework artificially bottlenecks the input frequency dimension $f$ into exactly 3 channels to simulate RGB video, which appears strictly necessary to load the Kinetics-400 weights. However, this raises an important question regarding the trade-off between preserving fine-grained physiological spectral features and utilizing natural video priors. To clarify whether the framework's performance is driven primarily by the VideoMAE architecture itself or the prior learning of Kinetics-400, could the authors address the following:
    - **Q1.1:** How exactly does the renderer compress the frequency dimension $f$ into just 3 channels, and what is the typical value of $f$ used in the experiments?
    - **Q1.2:** Could the authors perform an ablation experiment training the model from scratch (TFS) using a different embedding dimension $C \neq 3$ (e.g., $H \times W \times 8$ or $H \times W \times 16$) instead of forcing the $H \times W \times 3$ shape? The VideoMAE architecture's input would also be adapted to take in the new $C$.
    - **Q1.3:** How does the performance of a TFS model with a less restrictive bottleneck (higher $C$) compare to the 3-channel model initialized with Kinetics-400? Investigating this trade-off would significantly strengthen the paper.

2. Could the authors clarify how the mapping matrix $S$ is constructed universally across different datasets?

3. Is the renderer $R_v$ frozen during the VideoMAE self-supervised pre-training and downstream fine-tuning stages?

4. To better isolate the contribution of the proposed BSR architecture versus the massive non-EEG data prior (Kinetics-400), how does the "TFS" (Train From Scratch) model compare against the baselines (LaBraM, BIOT) in Tables 1 and 2? Is the state-of-the-art performance primarily driven by the mapping architecture itself, or by the pre-trained spatiotemporal dynamics of the video foundation model?

**Limitations:**

yes

**Strengths And Weaknesses:**

### Strengths

- The joint use of an autoencoder to map heterogeneous signals into a shared spatiotemporal grid, coupled with leveraging the learned prior of a spatiotemporal foundation model (VideoMAE) is truly novel. Using a pre-trained foundation model to inform the spatiotemporal dynamics of EEG data is a compelling initialization strategy.
- By evaluating performance on the full target dataset, the subset of channels seen during pre-training, and the completely disjoint set of unseen channels ($D_f^*$), the authors provide excellent empirical evidence that the model is learning extendable spatiotemporal dynamics rather than simply memorizing channel configurations.
- Handling the structural heterogeneity of varying electrode counts and placements is a major challenge for brain foundation models. Formulating this issue as mapping to a shared spatial grid provides a highly significant and scalable solution to cross-dataset EEG pre-training.
- The proposed method achieves considerable performance margins over very strong recent baselines (e.g., LaBraM, BIOT). (Though it would have been interesting to also compare with REVE (Ouahidi et al.), which the authors cite, as they similarly learn a shared coordinate space for montages).

### Weaknesses

- The widespread use of the term "video" is misleading. The encoded space from the autoencoder is simply a spatiotemporal tensor ($H \times W \times C$). There is nothing inherently "video" or "pixel"-like about this space. The present work seems instead to be exploring the use of pre-trained spatiotemporal foundation models as plausibile initializations for EEG data.
- The authors present "subject-level few-shot learning" and "cross-montage fine-tuning" as novel benchmark tasks. However, these settings are well-established in the domain adaptation and generalization literature. "Subject-level few-shot" corresponds directly to cross-subject transfer learning with per-subject calibration, and "cross-montage fine-tuning" is standard cross-device transfer addressing structural heterogeneity (Chen et al., 2025; Zhou et al., 2025).
- The core mechanism for cross-montage generalization relies on the mapping matrix $S$. However, the present manuscript does not explain how $S$ is universally constructed across completely different datasets. For cross-montage fine-tuning to function, the 2D grid $(h, w)$ must be anatomically aligned across all datasets, but the coordinate alignment process is left unexplained.
- The exact training dynamics of the pipeline are omitted. Specifically, it is unclear whether the video renderer $R_v$ is frozen after the initial Render-Reconstruct pre-training (Figure 3), or if it continues to be updated end-to-end during the VideoMAE self-supervised pre-training and downstream fine-tuning.

### Remarks

- The paper frames its primary contribution around "rendering EEG into videos." However, it appears the authors are well aware that the actual underlying mechanism is not the creation of a "video," but rather formatting EEG data into a structured spatiotemporal tensor to exploit the learned priors of a pre-trained foundation model (the Kinetics-400 weights). By leaning so heavily into the "video" metaphor, the manuscript obscures what is actually driving the state-of-the-art results. The narrative should be refocused to transparently communicate and analyze this transfer of spatiotemporal priors. I suggest the authors adjust the overarching narrative of the paper. Rather than framing the contribution as "transforming EEG into videos" (which is technically just mapping STFT features to a $H \times W \times C$ tensor), the paper would be much stronger if framed as *utilizing the learned spatiotemporal dynamics of a prior-trained foundation model for structured EEG representation learning*.
- In addition to the subject-level calibration setup, evaluating the model in a Leave-One-Subject-Out (LOSO) zero-shot setting would further strengthen the claims regarding the model's ability to generalize to unseen subjects without the need for calibration.
- There are instances of redundant citation formatting (e.g., "Authors et al. (Authors et al.)" instead of standard `\citet{}` formatting). Additionally, "that not appeared" should be corrected to "that did not appear".

---

> ### Author Rebuttal · Authors · 2026-03-31
>
> **W1&R1**
>
> We agree that the core contribution is formatting EEG as a spatiotemporal tensor to leverage pre-trained foundation model priors. The term Spectrum Video is kept for implementation reasons (e.g., save in .mp4 to save storage, using OpenCV API).
>
> **Q1**
>
> Q1.1: The Renderer consists of five transposed convolution layers, where the output channel count is determined by the final layer (Table 4 for detail). We use 3 channels as the typical value.
>
> Q1.2 & Q1.3: We setup the ablation experiment as follows:
> We conducted an ablation study under the following settings:
> 1.  We adopted the full‑dataset fine‑tuning paradigm to ensure sufficient training.
> 2.  We selected three different numbers of rendered channels: $N_{c}\in{1,3,8}$. For each setting, we compared TFS against loading Kinetics‑400 pretrained weights (with modifications to embed layers to adapt to 8‑channel model).
> 3.  For each $N_{c}$​, we pretrained the corresponding Renderer on the combined SEED and SHU‑MI datasets for 5,000 epochs.
>
> The results (of B-Acc) are summarized in the table below:
> |$N_{c}$|Pretrain|SEED|SHU-MI|
> |-|-|-|-|
> |1|TFS|0.3429|0.5008|
> |1|Load Kinetics|0.4918|0.5207|
> |3|TFS|0.3486|0.5025|
> |3|Load Kinetics|0.4878|0.5340|
> |8|TFS|0.3360|0.5015|
> |8|Load Kinetics|0.4995|0.5304|
>
> _Note: To save space, only avg is put._
>
> Findings:
> - The number of rendered channels has little effect on performance. This is likely because BSR does not compress information: the rendered spectrum image patch (size [32,32,$N_{c}$]​) provides a richer representation than the original spectral patch (size [1,1,101]).
> - TFS Models consistently underperform those initialized with Kinetics weights. This suggests that training VideoMAE from scratch is challenging, and data‑driven priors are crucial for strong performance.
>
> Conclusion: The number of rendered channels has a negligible impact on EEG representation quality, whereas pretraining from Kinetics‑400 yields substantial benefits. We therefore choose the 3‑channel configuration primarily for engineering convenience.
>
> **W3&Q2**
>
> The mapping matrix uses a fixed lookup table that assigns standard electrode names (10‑20/10‑10) to 2D grid coordinates. For each data, we create an empty tensor with shape [x,y,f] and place each electrode’s spectrum vector with shape [1,1,f] at its assigned (x,y) position, zero‑filling empty cells. A renderer then applies transposed convolution layers to convert each cell into a 32×32×3 image patch; patches are tiled into a final frame of size [32h,32w,3]. The renderer works on any grid layout (e.g., 5×5, 9×9) and can be pre‑trained once and applied to new datasets with standard electrode names.
>
> **W4&Q3**
>
> The status of the renderer can be summarized as follows: (1) During the initial Render‑Reconstruct pre‑training (Fig. 3), both the renderer and the reconstructor are trainable (not frozen). (2) After this stage, the reconstructor is discarded, and the renderer is frozen when it is used to generate spectrum videos for the subsequent VideoMAE training.
>
> **R2&R3**
> We agree that evaluating the model in LOSO settings would provide a stronger test, and we will add this in our future work. We also appreciate your careful reading regarding the citation formatting and the grammar error; these will be corrected in the revised manuscript.
>
> **Q4**
>
> We re-evaluated on few‑shot (Table 1) and full fine‑tuning (Table 2) using existing model checkpoints, comparing three variants of BSR‑VideoMAE:
> - TU2-TFS: trained from scratch on TUAB+TUEV (EEG only)
> - TU2+Kinetics: initialized from Kinetics‑400 then pretrained on TU2
> - Full: initialized from Kinetics‑400 then pretrained on 7 EEG datasets
>
> Table 1. Few-shot learning results (B-Acc).
> |Model|Pretrain|SEED|SEED-VII|SHU-MI|BCIC|
> |-|-|-|-|-|-|
> |VideoMAE|TU2-TFS|0.4890|0.1896|0.5764|0.3487|
> |VideoMAE|TU2+Kinetics|0.5282|0.2000|0.5986|0.3244|
> |VideoMAE|Full|0.5815|0.2054|0.5959|0.3275|
> |BIOT |\ |0.3677|0.1928|0.5456|0.2735|
> |LaBraM|Official|0.5263*|0.1554|0.5415|0.2836|
>
> Table 2. Full fine-tuning results (B-Acc).
> |Model|Pretrain|SEED-V|SHU-MI|
> |-|-|-|-|
> |VideoMAE|TU2 (TFS)|0.2644|0.5583|
> |VideoMAE|TU2 (+Kinetics)|0.2803|0.6182|
> |VideoMAE|Full (+Kinetics)|0.3155|0.6608|
> |BIOT|\ |0.2719|0.5614|
> |LaBraM|Official|0.2410|0.6421|
>
> _Note: LaBraM’s pre‑training includes SEED (see their Appendix D), so its SEED result is marked with * and only for reference._
>
> Findings:
> - In few‑shot, even the TU2 (TFS) model substantially outperforms baselines on all datasets. Gain on Kinetics initialization and larger pre‑training is incremental.
> - In full‑dataset fine‑tuning, larger‑scale pre‑training (Full) yields clear gains, as expected.
>
> Conclusion:  The SOTA performance of BSR‑VideoMAE in few‑shot is mainly driven by the BSR rendering framework, while in full fine‑tuning mainly benefit from the full pre‑training. Kinetics‑400 serves as a useful but non‑essential initialization.

---

> > ### Author Rebuttal · Reviewer_29UB · 2026-04-01
> >
> > I thank the authors for the detailed rebuttal and for promptly running the requested ablation experiments. The new comparisons clearly show the highly interesting result that spatiotemporal priors learned from natural video data can effectively be transferred to other spatiotemporal signals such as EEG. The results also demonstrate that the framework works with a varying number of rendered channels, making the practical choice of three channels perfectly fine. Because all of my main concerns including the specific technical questions regarding the mapping matrix and training dynamics were adequately addressed, I will raise my score to Accept given that these ablation results, implementation details, and the reframing of the evaluation settings as standard domain adaptation paradigms are all reflected in the revised manuscript.

---

> > > ### Author Response · Authors · 2026-04-02
> > >
> > > Dear Reviewer 29UB,
> > >
> > > Thank you for the thorough and constructive feedback, and for your positive recognition of our work. These suggestions have strengthened the experimental results and conclusions in this study, and are also very helpful for guiding our future research.
> > >
> > > Best regards, Authors of Paper 20779

---

### Official Review · Reviewer_FqGn · 2026-03-09

**Soundness:** 2
**Presentation:** 2
**Significance:** 2
**Originality:** 1
**Overall Recommendation:** 3
**Confidence:** 3

**Summary:**

The paper proposes a method called Brain Signal Rendering (BSR), which maps the spectral features of EEG into image sequences according to the spatial coordinates of electrodes, forming "spectrum videos". It then employs VideoMAE for self-supervised pre-training and fine-tuning on multiple downstream tasks. However, the novelty and contribution are somewhat limited.

**Compliance With Llm Reviewing Policy:**

Affirmed.

**Final Justification:**

Thank you for the author’s further response. The explanation that the discrepancy with the baseline paper is due to preprocessing and other factors raises concerns about fair comparison, as it may involve changes that deviate from the original method. Moreover, the overall comparison with the baseline remains insufficient. Regarding novelty, the author has not provided a more convincing rebuttal. Based on a reassessment, I have decided to adjust the score slightly.

**Key Questions For Authors:**

1. Does the cross-montage evaluation consider the influence of different referencing schemes (e.g., bipolar montages vs. referential montages)?
2.  Is masking also applied in the temporal dimension? Does the model account for the temporal dependencies inherent in EEG?
3.  How does the model perform on full-dataset fine-tuning tasks (e.g., on FACED，SEED, and SHU-MI) compared to methods like CBraMod and CSBrain? Are there performance gains?

**Limitations:**

yes

**Strengths And Weaknesses:**

*Strengths:*

1. The method considers the transferability across different devices and montage settings, which has rarely been systematically evaluated in existing EEG foundation model research.
2. By spatializing time–frequency spectra into video format based on electrode coordinates, BSR enables compatibility with video foundation models.

*Weaknesses:*

1. The core idea of mapping EEG spectra to 2D images according to electrode coordinates has been explored in prior work (e.g., [1]), limiting the novelty of the proposed BSR method.
2. The so-called “subject-level few-shot learning” treats each subject as a task with limited samples for fine-tuning. This essentially falls under within-subject or subject-independent evaluation, not the conventional definition of few-shot learning (which typically refers to learning from few examples per class). The terminology may mislead readers.
3. The experimental comparisons are not comprehensive enough; recent state-of-the-art methods such as CBraMod and CSBrain are not included.
4. The paper claims to adopt VideoMAE for pre-training but lacks details on critical aspects like the masking strategy. The sparsity of EEG channels in the 2D projection may behave differently from natural videos, yet this is not addressed.
5. Some writing issues: abbreviations such as “BL”, “FT-Ch”, “PT-Ch”, and “only PT-Ch” are used without explicit definition, imposing an unnecessary cognitive burden on readers.

[1] Yang L, Song Y, Jia X, et al. Two-branch 3D convolutional neural network for motor imagery EEG decoding[J]. Journal of Neural Engineering, 2021, 18(4): 0460c7.

---

> ### Author Rebuttal · Authors · 2026-03-31
>
> **W1**
>
> We acknowledge the interesting EEG-to-image approach in [1] and will cite it in our related works. However, that method rearranges raw EEG into a 3D tensor via spatial interpolation, which differs substantially from our BSR framework. In contrast, we propose a data-driven rendering that transforms time–frequency spectra into spectrum videos, enabling direct use of pretrained video foundation models. We hope this clarifies the novelty and unique contribution of our method.
>
> [1] Yang L, Song Y, Jia X, et al. Two-branch 3D convolutional neural network for motor imagery EEG decoding[J]. Journal of Neural Engineering, 2021, 18(4): 0460c7.
>
> **Q1**
>
> Thank you for this insightful question. Our current cross‑montage evaluation focuses on generalization across electrode spatial layouts, not explicitly on reference schemes. The datasets used (TUAB, TUEV, SEED, BCIC‑IV‑2a, etc.) follow their original preprocessing pipelines with different references. BSR spatializes electrodes by their physical coordinates; in theory, re‑referencing is a linear transformation of the measurement basis and does not alter spatial topology. We plan to add reference‑alignment and bipolar generalization experiments in future work.
>
> **W4 & Q2**
>
> Yes, temporal masking is applied. VideoMAE uses joint space‑time attention and masks 3D patches of size (2 frames × 16 × 16). This forces the model to reconstruct masked spatiotemporal cubes from visible context, explicitly learning temporal dynamics. The joint attention mechanism captures long‑range temporal dependencies and cross‑correlations between time and space. In BSR, the time axis corresponds to successive STFT windows, so temporal masking encourages the model to infer missing spectral frames from neighboring ones.
>
> **W3 & Q3**
>
> Thank you for the valuable feedback. In our initial selection of baselines, we primarily focused on spectrum-based methods (e.g., FFCL, ContraWR, C-Trans, BIOT, LaBraM) in order to control for the influence of the FFT transform itself (see the discussion on P300 in Appendix F of EEGPT [1]), thereby enabling a cleaner validation of the effectiveness of our proposed Brain Signal Rendering (BSR) framework. However, as the reviewer rightly pointed out, to fully demonstrate the superiority of BSR-VideoMAE, it is necessary to compare it with recent state-of-the-art models such as CBraMod and CSBrain.
>
> To this end, we have conducted supplementary experiments comparing BSR-VideoMAE with CBraMod and CSBrain on full-dataset fine-tuning tasks using the SEED and SHU-MI datasets. The results are presented in the following tables.
>
> Table1. Full-dataset fine-tuning results on SEED.
> |  | B-ACC | cohen | f1 |
> |--|--|--|--|
> | CBraMod | 0.4985 ± 0.0351 | 0.2366 ± 0.0507 | 0.4540 ± 0.0436 |
> | CSBrain | 0.4987 ± 0.0043 | 0.2374 ± 0.0076 | 0.4689 ± 0.0135 |
> | BSR-VideoMAE | **0.5570** ± 0.0022 | **0.3239** ± 0.0027 | **0.5247** ± 0.0032 |
>
> Table2. Full-dataset fine-tuning results on SHU-MI.
> |  | B-ACC | roc_auc | pr_auc |
> |--|--|--|--|
> | CBraMod | 0.6550 ± 0.0089 | 0.7175 ± 0.0101 | 0.7092 ± 0.0137 |
> | CSBrain | 0.6329 ± 0.0037 | 0.6983 ± 0.0050 | 0.7003 ± 0.0034 |
> | BSR-VideoMAE | **0.6608** ± 0.0068 | **0.7324** ± 0.0014 | **0.7396** ± 0.0032 |
>
> From the above results, it can be observed that BSR-VideoMAE consistently and significantly outperforms CBraMod and CSBrain across all metrics on both the SEED and SHU-MI datasets. These results further demonstrate that the BSR framework, by explicitly modeling electrode spatial geometry and spectral temporal dynamics, learns more generalizable representations.
>
> [1] Wang, Guagnyu, et al. "Eegpt: Pretrained transformer for universal and reliable representation of eeg signals." _Advances in Neural Information Processing Systems_ 37 (2024): 39249-39280.

---

> > ### Author Rebuttal · Reviewer_FqGn · 2026-04-01
> >
> > Thank you greatly for your response to our questions and for the additional experiments. However, we still have a question: why is there a discrepancy between the baseline accuracy reported in the paper and the results reported in the original papers of these baseline methods?
> >
> > **New**：Thank you for the author’s further response. The explanation that the discrepancy with the baseline paper is due to preprocessing and other factors raises concerns about fair comparison, as it may involve changes that deviate from the original method. Moreover, the overall comparison with the baseline remains insufficient. Regarding novelty, the author has not provided a more convincing rebuttal. Based on a reassessment, I have decided to adjust the score slightly.

---

> > > ### Author Response · Authors · 2026-04-01
> > >
> > > Thanks for the follow-up question. Generally, discrepancy (of the full-dataset fine-tuning tasks) between the baseline accuracy reported in our paper and the results reported in the original papers of baseline methods came from both difference in pre-processing and software/hardware difference [1].
> > >
> > > In particular, the length of all data in SEED, SEED-V and SEED-VII datasets are split into **10 seconds** in our study, which might be different from settings in CBraMod (which utilizes 1 second segment). This may be the primary reason for the significant discrepancy in SEED-V full-dataset fine-tuning between our results and those of the baseline methods.
> > >
> > > [1] PyTorch. (2024). _Reproducibility_. https://docs.pytorch.org/docs/stable/notes/randomness.html

---

### Decision · Program_Chairs · 2026-04-30

**Decision:**

Accept (regular)

**Comment:**

Three reviewers recommend ratings of 3, 4, and 5. The paper proposes Brain Signal Rendering (BSR), a framework that transforms EEG signals into geometry-aware Spectrum Videos and leverages VideoMAE pretrained on Kinetics-400 for self-supervised learning. Reviewer 29UB highlights that "the joint use of an autoencoder to map heterogeneous signals into a shared spatiotemporal grid, coupled with leveraging the learned prior of a spatiotemporal foundation model is truly novel." Reviewer nBgg appreciates the "original perspective" of "reframing electrode heterogeneity as a physical projection problem." Both reviewers recognize the cross-montage fine-tuning protocol as a valuable contribution for benchmarking EEG model generalization.

The primary concerns center on novelty and baseline fairness. Reviewer FqGn notes that "the core idea of mapping EEG spectra to 2D images according to electrode coordinates has been explored in prior work," citing Yang et al. (2021). Reviewer FqGn also raises that "the experimental comparisons are not comprehensive enough" and questions the discrepancy between reported baseline results and original papers. Reviewer 29UB acknowledges the "widespread use of the term video is misleading" and that "subject-level few-shot learning" terminology "corresponds directly to cross-subject transfer learning with per-subject calibration" in established literature. Missing details include the masking strategy and how phase information loss affects representation quality.

The rebuttal provided more ablation studies that addressed most concerns of reveiwers. Reviewer 29UB states "the authors provided a thorough rebuttal with the exact ablation studies requested" showing that "the natural video prior is indeed a major driver of the downstream performance." Reviewer nBgg confirms "the authors' response has resolved the issues I raised" and increased their score. Reviewer FqGn raised their score slightly but maintains that "the explanation that the discrepancy with the baseline paper is due to preprocessing raises concerns about fair comparison." The AC initiated discussion where Reviewer FqGn stated they "would not argue against acceptance, though my concerns remain," and Reviewer 29UB confirmed support while acknowledging FqGn's valid limitations.
Given that all reviewers raised their scores after rebuttal and Reviewer FqGn explicitly stated they would not argue against acceptance, the AC recommends acceptance, following reviewer majority opinion.

On balance, AC agrees with positive points raised by all reviewers which outweigh the negative points. The authors are strongly encouraged to include the additional reviewer recommendations, experiments from rebuttal and clarifications in the camera-ready version. Specifically, the authors should incorporate the Kinetics-400 ablation studies comparing TFS versus pretrained weights, add comparisions with CBraMod and CSBrain, clarify that "subject-level few-shot" corresponds to standard cross-subject transfer with calibration, discuss the Yang et al. (2021) prior work, as well as work that is close to VideoMAE (the central method of the paper) and MAE variants, more explicilty, and document the preprocessing differences that cause baseline result discrepencies.